

# Probabilistic projections and past trends of sea level rise in Finland

Havu Pellikka[1,2], Milla M. Johansson[2], Maaria Nordman[1,3], and Kimmo Ruosteenoja[2]

[1]Department of Built Environment, School of Engineering, Aalto University, Espoo, Finland
[2]Finnish Meteorological Institute, Helsinki, Finland
[3]Finnish Geospatial Research Institute, National Land Survey of Finland, Masala, Finland

**Correspondence:** Havu Pellikka (havu.pellikka@aalto.fi)

**Abstract.** We explore past trends and future projections of mean sea level (MSL) on the Finnish coast, in the northeastern Baltic Sea, in 1901–2100. We decompose the relative MSL change into three components: regional sea level rise (SLR), postglacial land uplift, and the effect of changes in wind climate. Past trends of regional SLR can be calculated after subtracting the other two components from the MSL trends observed by tide gauges, as the land uplift rates obtained from the semi-empirical model NKG2016LU are independent of tide gauge observations. According to the results, local absolute SLR trends are close to global mean rates. To construct future projections, we combine an ensemble of global SLR projections in a probabilistic framework. In addition, we use climate model results to estimate future changes in wind climate and their effect on MSL in the semi-enclosed Baltic Sea. This yields probability distributions of MSL change for three scenarios representing different future emission pathways. Spatial variations in the MSL projections result primarily from different local land uplift rates: under the medium emission scenario RCP4.5/SSP2-4.5, for example, the projected MSL change (5 to 95% range) over the 21st century varies from −28 (−54 to 24) cm in the Bothnian Bay to 31 (5 to 83) cm in the eastern Gulf of Finland.

## 1 Introduction

Sea level rise (SLR) is an existential threat to many coastal communities worldwide, but the magnitude and rate of future SLR are still shrouded in uncertainty. Over the last decade, advances in sea level projections have shifted the focus from global to regional assessments (e.g. Slangen et al., 2014; Kopp et al., 2014; Grinsted et al., 2015; Fox-Kemper et al., 2021) and from best estimates with uncertainty ranges to full probability distributions (Jevrejeva et al., 2014; Grinsted et al., 2015; Goodwin et al., 2017; Le Bars et al., 2017). A probability distribution of expected local mean sea level change allows different risk levels to be determined for different operations and types of infrastructure. Nuclear power plants, for example, need to be protected against much more unlikely risks than conventional buildings.

This paper examines past and future mean sea level (MSL) on the Finnish coast, located in the northeastern Baltic Sea (Fig. 1). The Baltic Sea is a semi-enclosed basin with a narrow ocean connection and a characteristic sea level behaviour that differs from open oceans. Wind conditions create decadal variations in sea level on the northeastern Baltic coasts, in Finland and Estonia (Johansson et al., 2014; Suursaar et al., 2006). Moreover, Fennoscandia is an area of substantial postglacial land uplift, as the Glacial Isostatic Adjustment (GIA) following the last ice age is still an ongoing process (Poutanen and Steffen, 2014). Land uplift, in particular, causes the MSL change in the study area to deviate markedly from the global mean.

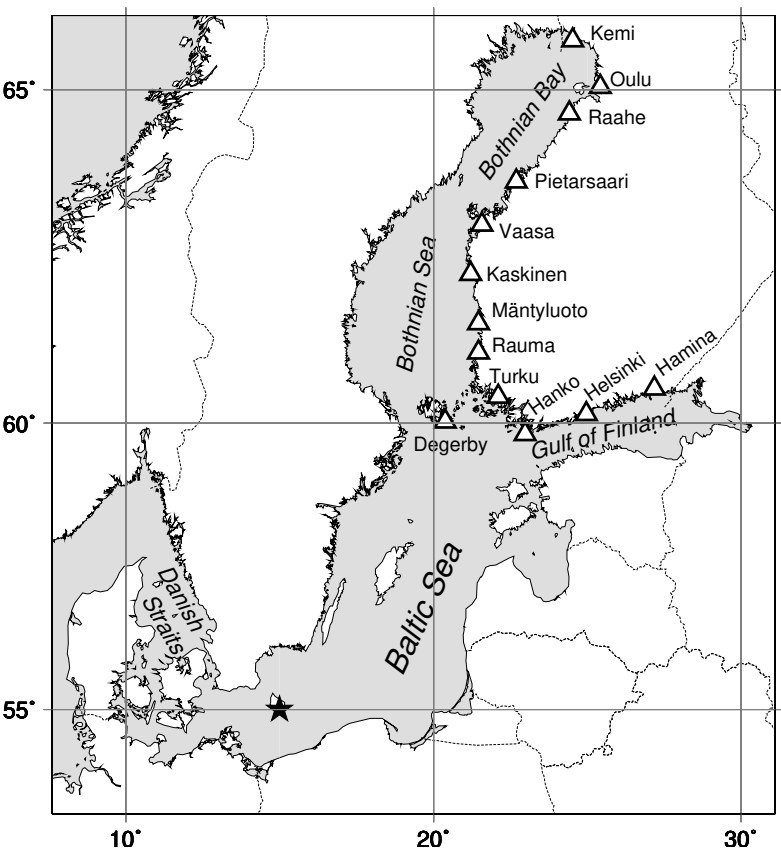

**Figure 1.** Map of the Baltic Sea region, with Finnish tide gauges marked with triangles. The star denotes the point (55°N, 15°E) with the best correlation between the zonal geostrophic wind speed component and mean sea level on the Finnish coast.

Despite major advances in the understanding of global mean sea level rise (GMSLR) and its causes, it has proved very difficult to nail down the upper end of the projection range. Upper limits of published GMSLR estimates have varied widely over the past 40 years when modern sea level projections have been available, and the total range of estimates in the literature has expanded rather than contracted (Garner et al., 2018). There is no consensus about the upper extreme tail of the probability

5   distribution. This is mainly because of poorly understood instability mechanisms of marine ice sheets – continental ice resting on ground below sea level and in direct contact with ocean, mainly found in West Antarctica (Fox-Kemper et al., 2021). Currently, one of the main disputes concerns the so-called Marine Ice Cliff Instability (MICI) hypothesis, the structural failure of ice cliffs after the loss of buttressing by floating ice shelves (DeConto and Pollard, 2016; Edwards et al., 2019; DeConto et al., 2021). Incorporating ice cliff failure in ice sheet models increases the Antarctic SLR contribution considerably, especially

10  after 2100 (Fox-Kemper et al., 2021).

The problem for coastal planners and other end users of SLR projections is that there are numerous projections available, which differ considerably from each other. The Intergovernmental Panel on Climate Change (IPCC) Assessment Reports



provide an authoritative overview of the state of climate science, and this information is widely used as a basis for national SLR assessments, e.g. in Norway (Simpson et al., 2015) and Sweden (Hieronymus and Kalén, 2020). A survey covering 32 European countries found IPCC to be the primary source of information for SLR planning (McEvoy et al., 2021). There are possible pitfalls in relying solely on IPCC projections, however, especially in sectors where risk aversion is critical. The consensus-based

approach of IPCC tends to produce rather conservative projections, and the upper limits of the IPCC projections have been consistently lower than the upper limits reported in individual studies (Garner et al., 2018). The Fifth Assessment Report (AR5, 2013) was criticized for disregarding low-confidence information and omitting the upper tail of the probability distribution from the numerical SLR projections (e.g. Hinkel et al. 2015; Siegert et al. 2020). In the recently published Sixth Assessment Report (AR6, 2021), IPCC for the first time provides local projections spanning the whole probability range. In addition, the marine

ice sheet uncertainty has been incorporated as a separate "low-likelihood, high-impact" storyline.

As Behar et al. (2017) point out, there has been a need to use multiple analyses and probability distributions in adaptation planning. In Finland, the approach chosen in coastal management and high-risk applications, such as the safety analyses of coastal nuclear power plants (Jylhä et al., 2018), has been to draw from a wide body of research in an attempt to create local SLR projections covering the full probability range (Johansson et al., 2014; Pellikka et al., 2018). In this study, we build on

previous work to calculate projections of mean sea level change in Finland. It has become timely to update earlier national projections after the publication of AR6 and other recent studies. Even though IPCC now provides localized projections, it is still worthwhile to use regional models and perform local analyses to improve local projections. As a small inland sea with a narrow ocean connection, Baltic Sea and its water exchange with the Atlantic Ocean is not well represented in global climate models. Moreover, the probability distributions presented in Johansson et al. (2014) and Pellikka et al. (2018) do not

differentiate between future greenhouse gas emission pathways. Merging all uncertainties into a single probability distribution was a choice based on practical needs, but this approach leads to a loss of information on how climate policies and other societal changes affect future SLR outcomes.

The Finnish tide gauge network has been measuring sea levels on the Finnish coast for over a century. Historically, land uplift has overruled sea level rise, causing declining trends in relative sea level and expansion of land area. It has, however,

been difficult to separate the different drivers of MSL change, as estimates of land uplift have partially relied on tide gauge observations. In recent years, the development of geodetic measurement techniques, such as Global Navigation Satellite System (GNSS) observations, has enabled independent estimates of land uplift rates. The current land uplift model for Fennoscandia, NKG2016LU (Vestøl et al., 2019), has already been used to estimate the rates of absolute SLR in the Baltic Sea by subtracting land uplift from observed relative sea level trends (Suursaar and Kall, 2018; Madsen et al., 2019; Passaro et al., 2021).

We use the long time series of Finnish tide gauge data to analyze past trends in MSL and its components. This analysis contributes to the understanding of regional SLR trends in the Baltic Sea, which has already been addressed in a number of studies. Using satellite altimetry, Stramska and Chudziak (2013) reported a trend of 3.3 mm a$^{-1}$ over 1992–2012, while Madsen et al. (2019) arrived at 3.4 mm a$^{-1}$ over 1993–2014, in accordance with the GMSLR trend. Over the 20th century, the Baltic Sea average was 1.3 mm a$^{-1}$ according to Madsen et al. (2019), but Suursaar and Kall (2018) reported 2.5 mm a$^{-1}$

in 1901–2010 based on Estonian tide gauge data. The difference may be explained by significant spatial variability within the





Baltic Sea, with larger rates in northern and eastern parts of the basin and lower rates in the southwestern areas (Gräwe et al., 2019; Madsen et al., 2019; Passaro et al., 2021). This pattern is attributed to wind and air pressure changes (Gräwe et al., 2019).

In this study, we utilize tide gauge data, modelling results, and SLR projections from scientific literature to analyze past and future MSL in Finland. The following research questions summarize the aims of this paper:

1. What was the rate of absolute sea level rise in the northeastern Baltic Sea over the past century and the satellite altimetry era?

2. What are the expected changes in mean sea level in Finland by 2100?

3. How do the probability distributions of SLR provided by IPCC AR6 compare with other recently published projections, and can they satisfy the needs of high-risk coastal management?

While the results of this study are mostly applicable for national planning purposes in Finland, the third question is not confined to the study area, and our analysis of the probability distribution of GMSLR may be of interest to coastal planners in other countries, as well.

## 2   Data and methods

### 2.1   Mean sea level components on the Finnish coast

Throughout this paper, mean sea level (MSL) refers to long-term sea level relative to land, while sea level rise (SLR) refers to the absolute sea level change relative to the centre of the Earth. SLR is a global process driven by thermal expansion of seawater and melting of glaciers and ice sheets, but local and regional SLR rates may deviate from the global mean. In the terminology of this paper, SLR is one component of the local MSL change. The MSL change in the study area may be positive or negative, depending on whether the land uplift is strong enough to overrule SLR.

We construct probability distributions of future MSL in Finland by combining three components: regional SLR, post-glacial land uplift, and the effect of changes in wind climate. Johansson et al. (2014) have shown that observed annual mean sea levels on the Finnish coast can be reproduced with good accuracy by computing the sum of these three components. More formally,

$$h_m(i,t) = r(i,t,t_0) - d(i)(t - t_0) + w(i,t) + R(i) \tag{1}$$

where $h_m$ is the estimate of local MSL at the tide gauge $i$ and time $t$, $r(i,t,t_0)$ is the regional SLR relative to some reference
25 year $t_0$, $d$ is the rate of land uplift, $w$ is the wind-induced MSL component, and $R$ is a constant used to level $h_m$ in relation to the Finnish N2000 height reference system (Saaranen et al., 2009) and the MSL of the reference year $t_0$. All sea level data used in this analysis are taken from the database of the Finnish Meteorological Institute (FMI).

Our method is presented schematically in Fig. 2. To obtain a probability distribution of $r$, we use a set of GMSLR projections published over the last decade (Sect. 2.2), also considering regional anomalies (Sect. 2.3). Land uplift rates $d$ are based on



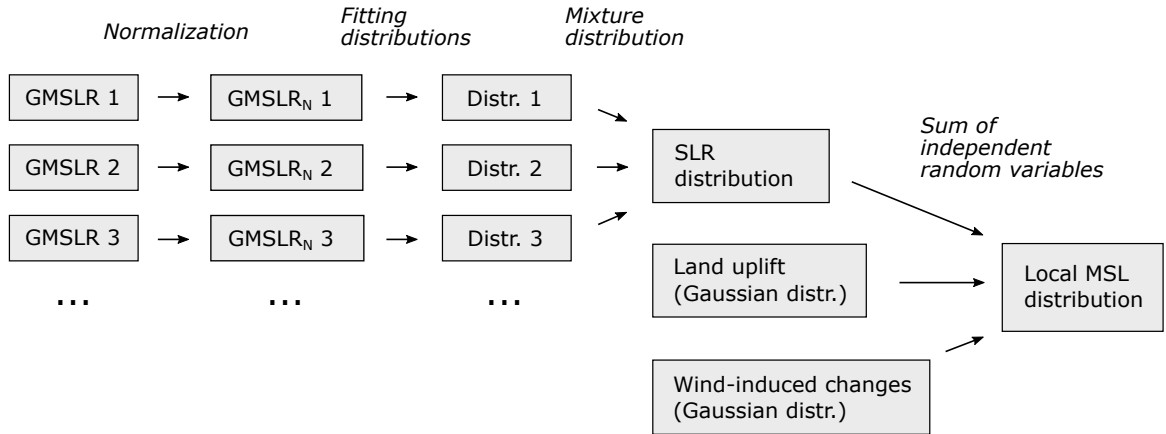

**Figure 2.** Schematic illustration of the method. A set of global mean sea level rise (GMSLR) projections is normalized to a common time span. Probability distributions are fitted to each projection and combined to produce a probability distribution of sea level rise (SLR), then combined with land uplift and wind-induced changes to produce local relative mean sea level (MSL) projections. This calculation is performed separately for different emission scenarios.

geodetic observations and modelling (Sect. 2.4). The wind-induced component $w$ is estimated using climate model projections of geostrophic wind (Sect. 2.5). Finally, we calculate the sum of the three MSL components to obtain a probability distribution of future MSL at a selected location on the Finnish coast. We treat the components as independent random variables, so the MSL distribution is the convolution of the distributions of the three components.

## 2.2 Probabilistic projections for sea level rise

To take into account the full range of potential future SLR, including low-probability, high-impact events such as marine ice sheet disintegration and controversial mechanisms such as MICI, we use a wide range of GMSLR projections published over the last decade. The projections are listed in Table 1. They are based on different approaches: i) *process-based models*, i.e. physical models that simulate the individual components contributing to SLR, including thermal expansion, ocean dynamics, and the melting and disintegration of land-based ice; ii) *semi-empirical models*, which project GMSLR indirectly by constructing a statistical relationship between global mean sea level and some other climate variable, e.g. the global mean temperature; and iii) *expert surveys* that assess the uncertainty in SLR projections by polling experts in the field and reviewing their understanding.

Scenarios of future greenhouse gas emissions play a crucial role in SLR projections. In IPCC AR6, the Representative Concentration Pathway (RCP) scenarios used in AR5 were replaced by a new set of emission scenarios called Shared Socioeconomic Pathways (SSP). The SSP and RCP scenarios are not directly comparable, but both are labelled by the level of radiative forcing reached in 2100. In the pre-AR6 literature, the most widely used RCP scenarios were RCP2.6, RCP4.5, and RCP8.5, which we also use in this paper since there is a sufficient number of GMSLR projections available for these scenar-





**Table 1.** Sea level rise projections used in this study: median values and 5–95% ranges (in parentheses). All values are in centimetres in 2100 relative to 1995–2014. The low emission scenario refers to RCP2.6/SSP1-2.6, medium to RCP4.5/SSP2-4.5, and high to RCP8.5/SSP5-8.5. AR6MC and AR6LC are the medium and low confidence projections of IPCC AR6 (Fox-Kemper et al., 2021), respectively.

| Publication | Low | Medium | High | Method |
|---|---|---|---|---|
| Jevrejeva et al. (2012) | 52 (32–77) | 68 (47–103) | 104 (76–157) | Semi-empirical model constrained by 300 years of tide gauge records and 1000-year reconstructions of radiative forcing |
| Jevrejeva et al. (2014) & Grinsted et al. (2015) | | | 78 (44–181) | Process-based + expert assessment of ice sheet contributions |
| Kopp et al. (2014) | 46 (28–80) | 56 (35–91) | 76 (52–118) | Process-based + expert assessment of ice sheet contributions |
| Kopp et al. (2016) | 36 (23–59) | 49 (32–83) | 74 (51–129) | Semi-empirical model calibrated to a 3000-year GMSL reconstruction |
| Mengel et al. (2016) | 36 (24–50) | 49 (33–72) | 81 (50–124) | Semi-empirical relations for each SLR component separately |
| Goodwin et al. (2017) | 54 (42–68) | 66 (52–80) | 87 (73–101) | Process-based thermosteric + semi-empirical ice melt contribution (their ensemble *ObsHist*) |
| Kopp et al. (2017) | 54 (25–96) | 89 (49–156) | 144 (91–241) | Process-based (as in Kopp et al., 2014, but replacing the Antarctic contribution with DeConto and Pollard, 2016) |
| Le Bars et al. (2017) | | 103 (34–173) | 180 (78–287) | Process-based (including DeConto and Pollard, 2016); their experiment DP16T |
| AR6MC | 44 (27–78) | 56 (37–94) | 77 (56–124) | Process-based, including medium-confidence processes |
| AR6LC | 45 (27–109) | 56 (36–117) | 88 (53–227) | Process-based + simulations incorporating MICI + structured expert judgement |

ios. The corresponding SSP scenarios are SSP1-2.6, SSP2-4.5, and SSP5-8.5. The SSP scenarios generally result in slightly stronger global warming than the corresponding RCP scenarios (Lee et al., 2021) and there are other differences as well, but for the purposes of this study, it is reasonable to group together the SSP and RCP scenarios with the same nominal radiative forcing. Hence, we construct probability distributions for three emission scenarios, which we call low (2.6), medium (4.5) and

5    high (8.5).

Our ensemble of GMSLR projections includes 8 members for the low emission scenario, 9 for the medium scenario, and 10 for the high scenario (Table 1). There are two sets of IPCC AR6 projections, medium confidence and low confidence (hereafter, AR6MC and AR6LC), which are included in our ensemble separately. In contrast to AR6MC, AR6LC incorporates information from structured expert judgment (Bamber et al., 2019) and ice sheet simulations (DeConto et al., 2021) that explore

10   the potential for rapid ice discharge from Antarctica through MICI. The probability of these high-end projections cannot be





robustly quantified; therefore, IPCC has decided to include them in a separate low confidence projection to serve the needs of stakeholders with a low risk tolerance. AR6MC and AR6LC differ from each other particularly under the high emission scenario.

IPCC AR6 sea level projections (Fox-Kemper et al., 2021) were downloaded from Physical Oceanography Distributed
Active Archive Center (PO.DAAC)[1] on Sep 29, 2021. Other data related to SLR projections are extracted from the tables or supplementary material of the referenced publications.

For each emission scenario, the ensemble of projections is transformed into a probability distribution of GMSLR in a three-step process (Fig. 2): First, the projections are normalized to a common baseline, 1995–2014. Second, a probability distribution is fitted to each projection. Third, the ensemble of distributions is combined to form a single probability distribution.

*1. Normalization.* All projections are normalized to a common baseline, 1995–2014, by assuming an initial GMSLR rate of $3.25 \pm 0.37$ mm a$^{-1}$ (Fox-Kemper et al., 2021 for 1995–2018) and constant acceleration from the initial rate. In practice, to normalize a projection having a baseline year $y_0$, we remove the rise realized between $y_0$ and 2005 after making a 2nd order fit to the projected SLR from $y_0$ to 2100 and the initial SLR rate.

*2. Fitting distributions.* To create an ensemble of continuous probability distributions, we fit distributions to the percentiles
of the published GMSLR projections. The number of percentiles available for the fit vary from three (median and the 5–95% range) to 31, depending on projection. There is no theoretical basis for choosing what type of distribution might best represent future SLR, but it is well known that the probability distributions of GMSLR are positively skewed with a fat upper tail. The tail represents the low-probability, high-impact scenarios that are hard to quantify. We experiment with three types of distributions that allow a positive skew: Weibull, Frechet, and skew normal. It turns out that no single distribution gives a good fit to all
projections, as the shape of the distribution varies between projections. We choose the best fit for each projection, defined as the smallest residual sum of squares (RSS) between the original data points and the fitted distribution in semi-log space ($x$, $log(1 - F)$, where $x$ is sea level rise and $F$ is the corresponding cumulative probability).

The fits to individual projections are shown in Fig. A1 for the high emission scenario. For IPCC AR6, which provides a large number of percentiles, we do not make a fit, but instead interpolate and extrapolate the distribution linearly in semi-log
space to make it continuous. Also, Jevrejeva et al. (2014) provide a continuous probability distribution, which we use as such (digitized from their Fig. 3).

*3. Combining distributions.* Finally, we calculate a mixture distribution to obtain the probability distribution of GMSLR for each emission scenario:

$$F(x) = \sum_{i=1}^{n} \lambda_i P_i(x) \qquad (2)$$

where $F$ is the cumulative distribution function (CDF) of GMSLR by 2100, $P_i$ are the CDFs of individual projections (the $n$ ensemble members), and $\lambda_i$ are weights attributed to each projection. Setting the weights is a matter of expert judgement. It is reasonable to assign different weights to the ensemble members, if there are plausible arguments for some projections

---

[1]https://podaac.jpl.nasa.gov/announcements/2021-08-09-Sea-level-projections-from-the-IPCC-6th-Assessment-Report





and projection methods to be more credible than others. We give a lower weight (0.5) to projections that incorporate MICI (Goodwin et al., 2017; Le Bars et al., 2017 and AR6LC) to factor in the low confidence in these projections. AR6MC is given a large weight (4) as it can be thought to represent the latest scientific consensus on SLR projections, and other projections are given equal weight (1). We explore different choices of weights, and the effects of the choices on the resulting projections, in
Sect. 3.2.

### 2.3    Considerations of regional SLR anomaly

Various processes create spatial inhomogeneity in SLR, including variations in ocean temperature, salinity, and circulation as well as gravitational, rotational, and solid Earth responses to ice melt. Each melting glacier produces a characteristic geographical pattern (fingerprint) of SLR, resulting from changes in Earth's gravity field and rotation as well as crustal deformation that
follow the reduced weight of the ice mass. Sea level rise is smaller than average in the proximity of the melting glacier or ice sheet, even negative up to the distance of ca. 2000 km, because of elastic uplift of the crust and the reduced gravitational pull of the ice mass (Mitrovica et al., 2011).

On the Finnish coast, the fingerprint effect makes the SLR contributions of the Greenland Ice Sheet and other northern glaciers notably smaller compared to the global mean. The contribution of Greenland is essentially zero in the Bothnian Bay
and ca. 15% of the global mean on the southern Finnish coast (Mitrovica et al., 2001; Kopp et al., 2014). On the other hand, according to global climate models, thermal expansion and ocean dynamical effects (hereafter referred to as the ocean component) produce a larger-than-average SLR contribution in the Baltic Sea. Pellikka et al. (2018) analyzed the ocean component from Coupled Model Intercomparison Project Phase 5 (CMIP5) models on the Finnish coast: median estimates for different emission scenarios were 20 to 60% higher compared to the global mean. This additional SLR was, however, overruled by the
reduction caused by glacier and ice sheet fingerprints, and the total SLR (excluding land uplift) was about 80% of the global mean in end-of-century projections.

In IPCC AR6, which uses the current CMIP6 climate models, the local ocean component on the Finnish coast is larger than previously estimated: almost twice the global mean in 2100. Fig. 3 shows the proportions of different SLR components – the ocean component, different glaciers and ice sheets, land water storage, and vertical land motion – in the AR6MC median
projections locally and globally. On the Finnish coast, the large ocean component compensates the reduced glacier components so that the absolute SLR is approximately the same as the global mean. Projections for Hanstholm, on the North Sea coast of Denmark, are plotted for comparison and show no notable difference. Therefore, the large ocean component is not a feature confined to the Baltic Sea basin, whose sea level dynamics the global models are not able to simulate well. The reason for the notably larger ocean component in CMIP6 compared to CMIP5 seems to be the increased dynamic sea level rise in the North
Atlantic, associated with a weakening of the Atlantic meridional overturning circulation (Lyu et al., 2020).

It is important to note that the conformity of global and local absolute SLR takes place only because two phenomena happen to cancel each other: the smaller-than-average glacial contribution and the larger-than-average ocean contribution. This may not be true for other, methodologically different projections, or projections with different time spans, as the relative proportions of the SLR contributors may be different. Localized projections based on essentially different methods (such as semi-empirical


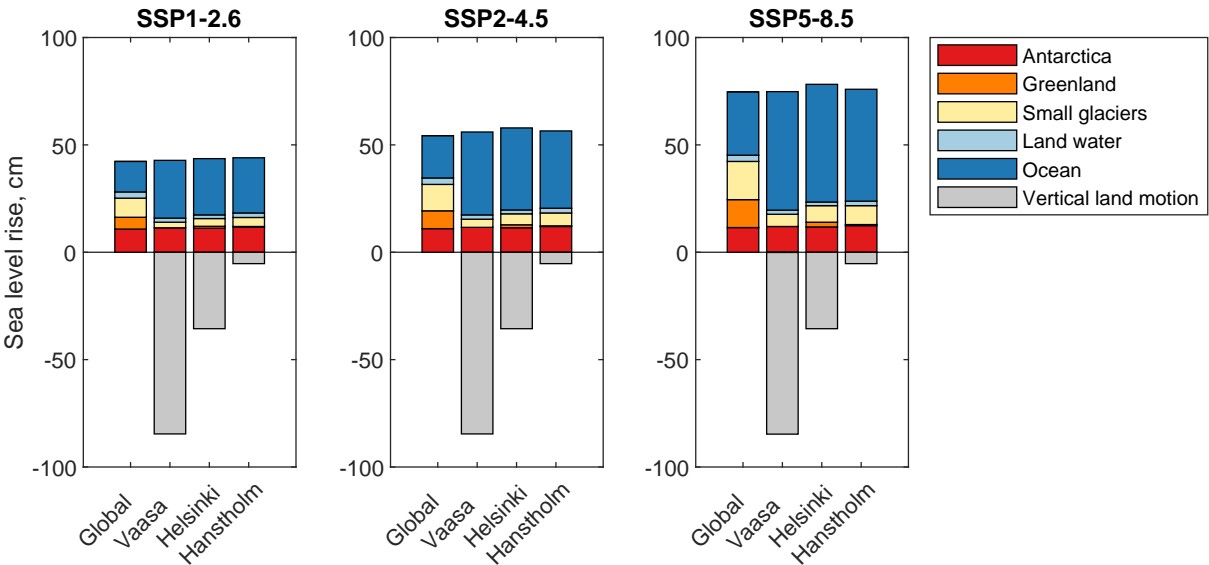

**Figure 3.** Median sea level rise projections of IPCC AR6 (medium confidence, 2100 relative to 1995–2014) divided into different components, for three emission scenarios. The ocean component includes thermal expansion and ocean dynamical effects. The global projection is shown together with three local projections: Vaasa and Helsinki on the Finnish coast, Hanstholm on the North Sea coast of Denmark.

modelling) are not available, however. To study projections with different time spans, we calculated the ratio of local to global SLR from AR6 median projections every 10 years from 2030 to 2150. In Helsinki, for example, the ratio declines from 1.2–1.4 in 2030 to 1–1.1 in 2100 and 0.9–1 in 2150 (Fig. 4), as the contribution of land ice melt increases and gradually tends to dominate the projected SLR.

Based on this analysis, we apply the GMSLR projections for 2100 to the Finnish coast as such, without regional adjustments. In other words, we approximate the regional SLR $r(i, t, t_0)$ in Eq. (1) with the global mean SLR, $g(t, t_0)$. Using a scaling approach similar to Pellikka et al. (2018) to convert GMSLR projections into regional ones would lead to small adjustments in the final projections, which are minor compared to uncertainties involved in such analysis. Therefore, we omit these adjustments for simplicity.

**2.4 Land uplift**

Vertical land motions (VLM) in the northern Baltic Sea area are dominated by Glacial Isostatic Adjustment (GIA). Slow crustal recovery from the pressure of the ice sheet that covered the area during the last glacial period causes post-glacial land uplift. On the Finnish coast, the rate of land uplift is of the same order of magnitude (3 to 9 mm a$^{-1}$) as the current rate of GMSLR (3.7 mm a$^{-1}$; Fox-Kemper et al. 2021). The uplift rate can be considered constant in time over the next few centuries (Poutanen
and Steffen, 2014).

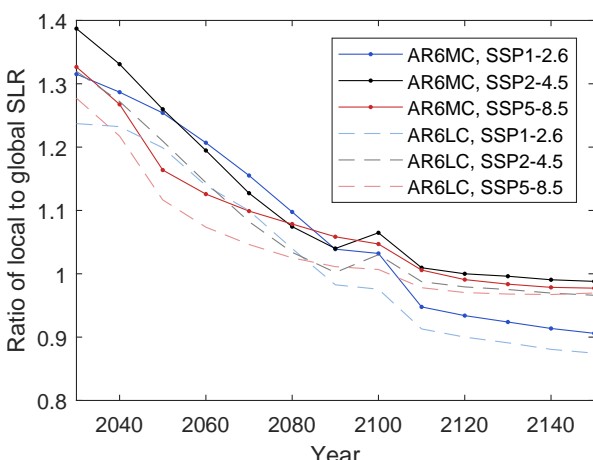

**Figure 4.** Ratio of local absolute sea level rise in Helsinki (without the effect of land uplift) to the global mean in IPCC AR6 median projections. Medium confidence projections (AR6MC) are plotted with dark lines, low confidence projections (AR6LC) with pale dashed lines, while colours indicate different emission scenarios.

We use land uplift rates calculated by NKG2016LU, a semi-empirical model of land uplift in the Fennoscandian area (Vestøl et al., 2019). The model has been computed in the Working Group of Geoid and Height Systems of the Nordic Geodetic Commission (NKG) and combines two parts: i) the observation-based part that uses geodetic observations of land uplift, namely levelling and GNSS time series in the area, and ii) the geophysical GIA model that supplements data to areas where

observations are sparse. The GIA model is fitted to the geodetic observations using Least Squares Collocation (LSC). The uncertainty estimate of the model is a combination of the observations' and the GIA model's uncertainty. The levelled version used in the current study gives the uplift relative to the geoid (Fig. 5). Land uplift rates, $d$, at the Finnish tide gauges are listed in Table 3 in Results (Sect. 3.1). We assume a normal (Gaussian) uncertainty distribution for the land uplift, using the 1 std uncertainty ranges of the model to fit the distributions.

The local projections of IPCC AR6 account for VLM by calculating background rates of mean sea level change from tide gauge data, using the statistical model of Kopp et al. (2014). The long-term background trend is scenario-independent and assumed constant over the projection period. This way, other drivers of long-term VLM such as tectonics, volcanism, and anthropogenic subsidence can be considered in addition to GIA. IPCC assigns low to medium confidence in VLM projections and points out that in many regions, more detailed regional analyses would be needed to produce higher-fidelity projections.

In Fennoscandia, VLM is dominated by GIA, which is a relatively well known and predictable process. Comparing the VLM rates of AR6 and the NKG2016LU model, the differences are small and result in a difference of a few centimetres at most over 100 years.

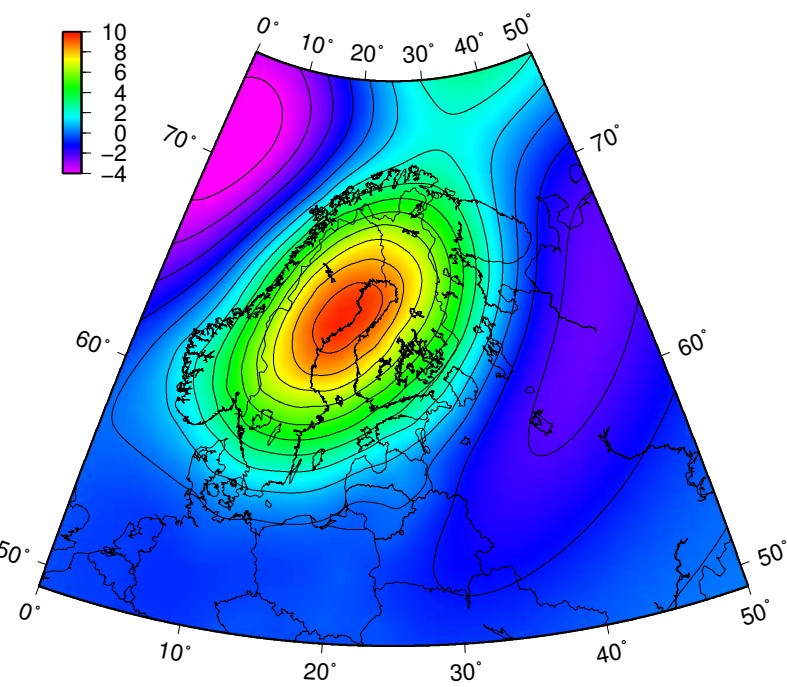

**Figure 5.** The present day land uplift (millimetres per year) over Fennoscandia relative to geoid given by the semi-empirical land uplift model NKG2016LU_lev.

## 2.5   Changes in wind climate

The wind-induced component $w$ (Eq. 1) is related to sea level dynamics within the Baltic Sea basin. The narrow and shallow Danish Straits are the only ocean connection of the Baltic Sea, and water exchange in the Straits plays an important role in Baltic sea level variations. The barotropic water exchange is governed by large-scale wind and air pressure patterns, which also

5    affect dynamical sea level within the Baltic Sea basin: persistent westerly winds push water through the straits and generate a slope in sea level, elevating sea levels on the eastern Baltic coast. Climatological changes in wind conditions can have a notable effect on long-term MSL on the Finnish coast (Johansson et al., 2014). We treat this component separately, as global climate models are too coarse to fully capture the water exchange in the Danish Straits and the dynamical sea level topography within the Baltic Sea.

10    Johansson et al. (2014) studied the correlation between the annual means of sea level on the Finnish coast and the zonal component of the geostrophic wind in the region. The best correlation was found with geostrophic winds calculated at the grid point 55°N, 15°E, close to the Danish Straits (Fig. 1). The zonal geostrophic wind $u_g$ at this grid point explains more than 80% of the interannual variability of sea level on the Finnish coast over the 20th century. Following Johansson et al. (2014),


we calculate $u_g$ from the Northern Hemisphere daily mean sea-level pressure fields for the years 1899–2018 (CISL Research Data Archive, 1979) according to the standard definition

$$u_g = -\frac{1}{f\rho}\frac{\partial p}{\partial y} \tag{3}$$

where $f$ is the latitude-dependent Coriolis parameter, $\rho$ the air density (calculated assuming a constant temperature of 283
5   K), and $\partial p/\partial y$ the pressure gradient in the north–south direction.

To project $u_g$ into the future, we study zonal geostrophic winds derived from the sea level pressure fields produced by the CMIP5 climate model runs (Taylor et al., 2012). We use results from 17 atmosphere–ocean general circulation models (AOGCMs), listed in Table A1. We select all models that i) are assessed by Luomaranta et al. (2014) as capable of adequately simulating the observed climate (temperature, precipitation) in northern Europe, and ii) provide information of daily sea level
pressure for all three emission scenarios.

From the daily mean sea level pressure fields of each model, we calculate $u_g$ using Eq. (3) (but instead of constant temperature, the modelled pressure and 2-m temperature fields are used to calculate air density). These are calculated to the middle of the grid points of each model and interpolated to the actual model grid. For more details, see Ruosteenoja et al. (2019). From the daily geostrophic winds, we calculate 20-year annual means for the historical reference period (1986–2005) and for the end
of the 21st century (2081–2100). Finally, the results are interpolated bilinearly to represent the point 55°N, 15°E.

The spread of the model-projected values of $u_g$ for the historical period (1986–2005) is large (Table A1), the averages for this 20-year period ranging from 0.0 to 4.9 m s$^{-1}$ among the 17-model ensemble, while the observational average (calculated from the CISL dataset) is 3.2 m s$^{-1}$. To compensate for this large inter-model variability, we use a delta-change approach: instead of using the projected values for the future time periods as such, we calculate the model-specific changes from the
historical period to the future period, and add these to the observational historical average of 3.2 m s$^{-1}$. This still results in a large spread of the model projections, ranging from 2.4 to 5.7 m s$^{-1}$ for 2081–2100. The means, maxima and minima, as well as standard deviations among the 17-model ensemble for each emission scenario are given in Table 2.

The wind-induced sea level component is then calculated as $w(i,t) = p_i u_g(t)$, where $p_i$ are regression coefficients between $u_g$ and the detrended annual mean sea levels at the tide gauge $i$, calculated from 20th century observations. For more details,
see Johansson et al. (2014). The regression coefficients range between 6.8 to 7.8 cm (m s$^{-1}$)$^{-1}$. The probability distribution for future $w(i,t)$ is a product of two normal (Gaussian) distributions, whose standard deviations are the regression fit uncertainty of $p_i$ and the standard deviation of the 17-model ensemble of $u_g$.

## 2.6   Past regional SLR trends

The land uplift rates given by the NKG2016LU model are solely based on geodetic observations. Therefore, they are inde-
pendent of tide gauge observations and other sea-level related information. This allows us to estimate the rate of SLR on the Finnish coast over the 20th century using the three-component model of Eq. (1), the unknown variable being the regional SLR $r$. For all Finnish tide gauges, we take the observed annual mean sea levels ($h_m$) and subtract the wind-induced component ($w$)





**Table 2.** Projections of zonal geostrophic wind in 2081–2100 at 55°N, 15°E: means, maxima, minima, and standard deviations (metres per second) among the 17-model ensemble under low (RCP2.6), medium (RCP4.5) and high (RCP8.5) emission scenarios. The values are calculated so that the projected change from the historical period 1986–2005 is added to the observational average of 3.2 m s$^{-1}$ over that period.

|      | Low  | Medium | High |
|------|------|--------|------|
| Mean | 3.29 | 3.46   | 3.96 |
| Max  | 4.26 | 4.35   | 5.68 |
| Min  | 2.40 | 2.75   | 2.61 |
| Std  | 0.57 | 0.49   | 0.80 |

and the land uplift ($d$) from the time series. The trend calculated from the residual corresponds to the local absolute SLR rate. Removing $w$ from the annual mean sea levels reduces the year-to-year variability; therefore, $r$ has narrower uncertainty ranges than $h_m$.

## 3 Results

### 3.1 Rate of past sea level rise

Past SLR trends for two time spans, $t_0$–2018 and 1993–2018, are shown in Table 3 for all Finnish tide gauges: $t_0$ is the beginning of the observational period and, depending on the site, varies between 1901–1933. The historical SLR trends ($t_0$–2018) on the Finnish coast range from 1.2 to 1.5 mm a$^{-1}$, in accordance with some recent estimates of the rate of the 20th century GMSLR (Hay et al., 2015; Dangendorf et al., 2017; Frederikse et al., 2020) but slightly lower than the AR6 estimate of $1.73 \pm 0.45$ mm a$^{-1}$ (Fox-Kemper et al., 2021). Local trends obtained for the satellite altimetry era, 1993–2018, range from 3.0 to 3.8 mm a$^{-1}$, in accordance with the global mean rate of $3.25 \pm 0.37$ mm a$^{-1}$ (Fox-Kemper et al., 2021).

### 3.2 Ensemble projection of GMSLR and the effect of weighting

We have combined 10 SLR projections published over the last decade (Table 1) to yield probability distributions of GMSLR for three emission scenarios. This analysis results in the following median estimates (5 to 95% ranges) of GMSLR in 2100 relative to 1995–2014: 45 (26 to 78) cm for the low emission scenario, 60 (36 to 111) cm for the medium scenario, and 85 (54 to 176) cm for the high scenario. Fig. 6 shows our probability distributions compared with the two IPCC AR6 projections, AR6MC and AR6LC, plotted as complementary cumulative distribution functions (1 – CDF) in semi-log scale so that differences in the upper tail are clearly visible. Interestingly, the combined distribution is nearly identical to AR6MC under the low emission scenario, but close to AR6LC under the medium scenario. Under the high scenario, the combined projection lies between AR6MC and AR6LC. The results are not drastically different even if the two AR6 projections are dropped out of the ensemble and we only use the 8 other projections to calculate the combined projection.





**Table 3.** Historical trends (millimetres per year) of observed mean sea level $h_m$ and its components: land uplift $d$, wind-induced component $w$, and regional sea level rise $r$. Error estimates are 1 std. The trends are calculated from the beginning of the observations ($t_0$) to 2018; the trend of regional sea level rise $r$ is also shown for 1993–2018.

| Station | $t_0$ | $\dot{h_m}$ | $-\dot{d}$ | $\dot{w}$ | $\dot{r}$ | $\dot{r}$ |
|---|---|---|---|---|---|---|
| | | | $t_0$–2018 | | | 1993–2018 |
| Kemi | 1923 | $-6.68 \pm 0.27$ | $-8.60 \pm 0.23$ | $0.64 \pm 0.27$ | $1.28 \pm 0.14$ | $3.76 \pm 0.90$ |
| Oulu | 1923 | $-6.26 \pm 0.27$ | $-8.30 \pm 0.20$ | $0.63 \pm 0.26$ | $1.36 \pm 0.13$ | $3.75 \pm 0.92$ |
| Raahe | 1923 | $-6.74 \pm 0.26$ | $-8.71 \pm 0.19$ | $0.64 \pm 0.27$ | $1.31 \pm 0.12$ | $3.40 \pm 0.89$ |
| Pietarsaari | 1922 | $-7.04 \pm 0.26$ | $-8.95 \pm 0.17$ | $0.63 \pm 0.26$ | $1.30 \pm 0.12$ | $3.59 \pm 0.81$ |
| Vaasa | 1922 | $-6.97 \pm 0.25$ | $-8.79 \pm 0.16$ | $0.60 \pm 0.25$ | $1.23 \pm 0.11$ | $3.54 \pm 0.76$ |
| Kaskinen | 1927 | $-6.29 \pm 0.28$ | $-8.31 \pm 0.17$ | $0.74 \pm 0.27$ | $1.35 \pm 0.13$ | $3.50 \pm 0.78$ |
| Mäntyluoto | 1926 | $-5.48 \pm 0.25$ | $-7.44 \pm 0.22$ | $0.67 \pm 0.26$ | $1.25 \pm 0.12$ | $3.44 \pm 0.77$ |
| Rauma | 1933 | $-4.53 \pm 0.29$ | $-6.81 \pm 0.17$ | $0.79 \pm 0.30$ | $1.48 \pm 0.13$ | $3.41 \pm 0.77$ |
| Turku | 1922 | $-3.56 \pm 0.24$ | $-5.41 \pm 0.17$ | $0.59 \pm 0.25$ | $1.29 \pm 0.11$ | $3.18 \pm 0.80$ |
| Degerby | 1924 | $-3.70 \pm 0.24$ | $-5.70 \pm 0.20$ | $0.58 \pm 0.24$ | $1.37 \pm 0.11$ | $3.04 \pm 0.78$ |
| Hanko | 1901 | $-2.51 \pm 0.17$ | $-4.16 \pm 0.19$ | $0.45 \pm 0.18$ | $1.22 \pm 0.08$ | $3.21 \pm 0.85$ |
| Helsinki | 1904 | $-1.88 \pm 0.19$ | $-3.69 \pm 0.18$ | $0.49 \pm 0.19$ | $1.32 \pm 0.08$ | $3.15 \pm 0.85$ |
| Hamina | 1929 | $-0.87 \pm 0.31$ | $-3.03 \pm 0.18$ | $0.81 \pm 0.31$ | $1.32 \pm 0.13$ | $3.50 \pm 0.88$ |

As long as unresolved questions around ice sheet instability remain, sea level projections aiming to cover the whole probability range unavoidably rely to some extent on educated guesses, or expert judgment. Expert judgment is involved in the individual GMSLR projections (Table 1). In our method, there is subjectivity in the choice of the type of distribution fitted, if there are multiple good fits, and the weights given to individual projections in the mixture (Eq. 2). Next, we examine how sensitive our results are to these choices.

As seen from Fig. A1, the three distribution types (Weibull, Frechet, and skew normal) diverge substantially in the extreme tail of the probability distribution. In some cases, it is clear which distribution provides the best fit, but often any one of them could be chosen, while for Kopp et al. (2017) no good fit is found. We have tested different fitting strategies, such as always choosing the most pessimistic or the most optimistic option when multiple good fits are available, and the differences in the resulting mixture distribution are small (max. 2 cm in the 99.9th percentile). We conclude that our method is robust to the choice of the distribution type at least up to the cumulative probability of 99.9%. Extrapolations beyond that are in any case very speculative, and we avoid presenting such results.

What matters more is the weight given to the various members of the projection ensemble when calculating the mixture, especially the weight given to projections that incorporate MICI. We experiment with different weighting strategies to see how much the results are affected by the weights:

1. All members are included with an equal weight.





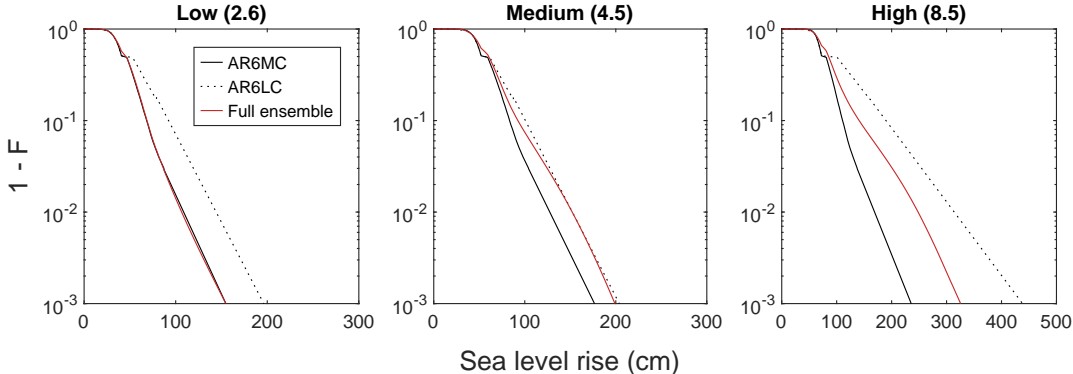

**Figure 6.** Complementary cumulative distribution functions (1 − F) of global mean sea level in 2100 relative to 1995–2014 under different emission scenarios. The red line denotes a weighted combination of 10 sea level rise projections, including the two IPCC AR6 projections, medium and low confidence (AR6MC and AR6LC, respectively). AR6MC and AR6LC are plotted separately for comparison.

2. AR6MC is given a 4-fold weight, as it can be considered to represent the latest scientific consensus on SLR projections.

3. Projections that incorporate MICI (Goodwin et al., 2017; Le Bars et al., 2017; and AR6LC) are given a lower weight (0.5) which reflects the low confidence in these projections. AR6MC is given a weight of 4 and the other members 1.

4. Projections that incorporate MICI are excluded from the ensemble, while AR6MC is given a weight of 4.

5   Increasing the weight of AR6MC somewhat lowers the final projection, but the difference is ca. 10 cm at most compared to the unweighted case. Unsurprisingly, lowering the weight of the MICI projections has a large effect on the upper tail of the distribution under higher emission scenarios. The 99.9th percentile is lowered by 0.15 m, 0.5 m, and 0.9 m under the low, medium and high scenarios, respectively, if MICI projections are dropped out compared to the unweighted case. For the final results, we choose the fitting strategy 3, which represents a middle road – we want to take into account the MICI uncertainty, but not give it too much weight because of the controversiality. By adjusting the weights, the projections could be tailored to specific purposes according to the risk tolerance of the application.

### 3.3   Future mean sea level on the Finnish coast

After adding the land uplift and the wind-induced component related to the sea level dynamics within the Baltic Sea basin, we obtain probability distributions of future MSL on the Finnish coast. The distributions of the three components and the total mean sea level change over the 21st century are shown in Fig. 7 for the medium emission scenario. Projections of MSL change are tabulated for selected locations in Table 4, while projections for all Finnish tide gauges are provided in the Appendix (Tables A2, A3, A4). The projected MSL values are also given in the Finnish N2000 height reference system (Saaranen et al., 2009). Mean sea level in N2000 over the reference period 1995–2014 was averaged from observed annual mean sea levels.




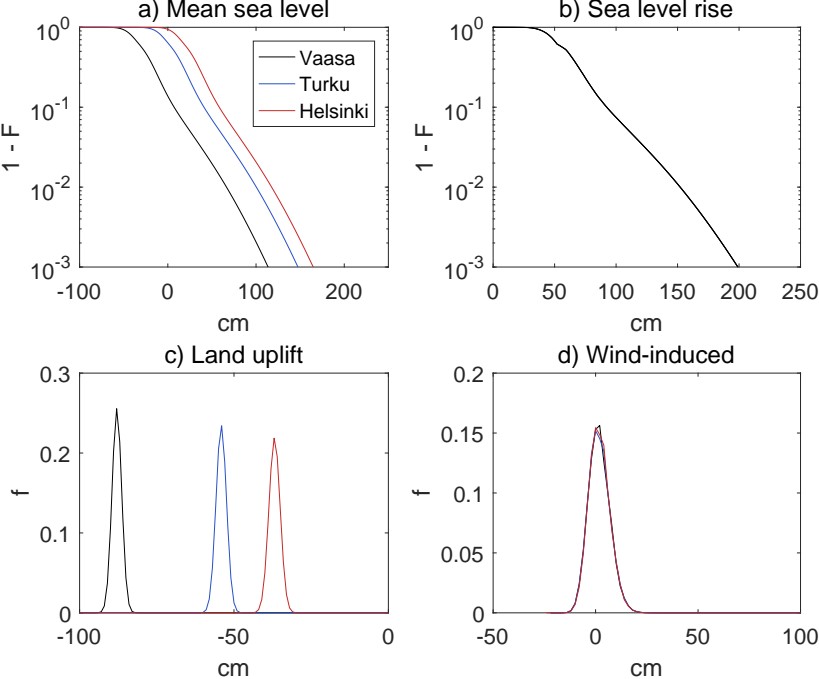

**Figure 7.** Probability distributions of projected mean sea level (a) and its components (b to d) in Vaasa, Turku, and Helsinki under the medium emission scenario (RCP4.5/SSP2-4.5), 2100 relative to 1995–2014. The distributions a) and b) are complementary cumulative distribution functions $(1 - F)$, while c) and d) are probability density functions (f).

**Table 4.** Mean sea level change (centimetres) from 1995–2014 to 2100 in some of the largest coastal cities in Finland: Oulu, Vaasa, Pori, Turku, and Helsinki. The global sea level rise over the same period, calculated as a weighted combination of 10 SLR projections, is shown for comparison. The 5th, 50th, 95th, and 99th percentiles are given for three emission scenarios: low (RCP2.6/SSP1-2.6), medium (RCP4.5/SSP2-4.5) and high (RCP8.5/SSP5-8.5).

|  | Low | | | | Medium | | | | High | | | |
|---|---|---|---|---|---|---|---|---|---|---|---|---|
| Station | 5% | 50% | 95% | 99% | 5% | 50% | 95% | 99% | 5% | 50% | 95% | 99% |
| Oulu | −59 | −37 | −2 | 25 | −47 | −21 | 30 | 72 | −27 | 8 | 99 | 171 |
| Vaasa | −64 | −42 | −8 | 20 | −52 | −26 | 25 | 67 | −31 | 3 | 94 | 166 |
| Mäntyluoto (Pori) | −50 | −28 | 6 | 34 | −39 | −13 | 39 | 80 | −18 | 16 | 107 | 180 |
| Turku | −30 | −8 | 26 | 54 | −18 | 7 | 59 | 100 | 2 | 36 | 128 | 200 |
| Helsinki | −13 | 9 | 43 | 71 | −1 | 25 | 76 | 118 | 20 | 54 | 145 | 217 |
| *Global* | *26* | *45* | *78* | *107* | *36* | *60* | *111* | *152* | *54* | *85* | *176* | *248* |

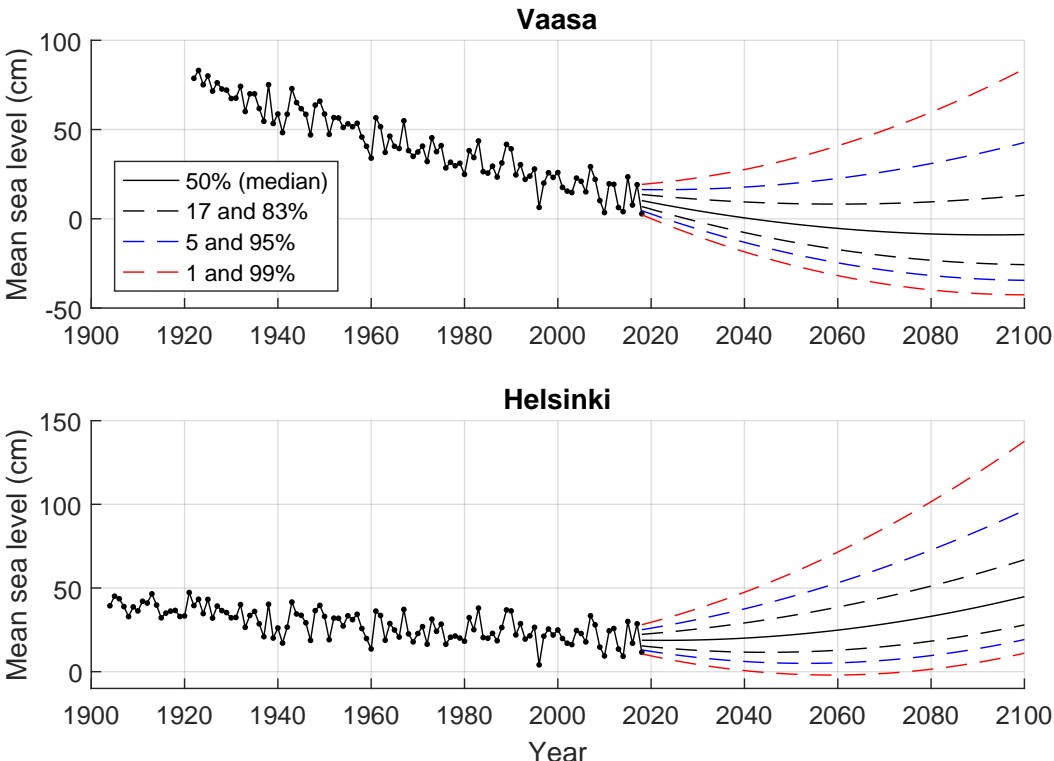

**Figure 8.** Observed annual mean sea levels and future projections under the medium emission scenario (RCP4.5/SSP2-4.5) in Vaasa, region of strong land uplift, and Helsinki, where the land uplift is weaker. The reference level is the Finnish N2000 height reference system.

Under the low emission scenario, substantial MSL rise is not expected on the Finnish coast over the 21st century: the median estimate ranges from a −43 cm decline in the Bothnian Bay to a 16 cm rise in the eastern Gulf of Finland (Table A2). However, according to the high-end (95%) estimate, a MSL rise of half a metre is possible on the southern Finnish coast. Under the medium scenario, a sea level decline of −20 to −30 cm is expected in the Bothnian Bay, whereas the Gulf of Finland shows a

5 MSL rise of 20 to 30 cm with an upper limit (95%) of 70 to 80 cm. The high emission scenario is the only one in which sea level rise is expected over the whole Finnish coastline, according to the median projection. The 95th percentiles are close to 1 m in the Bothnian Bay and 1.5 m in the Gulf of Finland.

Fig. 8 shows the observed and projected MSL change over time in Vaasa and Helsinki. The future time series are fitted to the 2100 projections assuming constant acceleration in the MSL trend. Vaasa is located in the region of strongest land uplift

10 (8.8 mm a$^{-1}$) while in Helsinki, the rate of land uplift is considerably weaker (3.7 mm a$^{-1}$). This is reflected in the past and future MSL trends. In Helsinki, the historical decline of mean sea level is about to reverse, and MSL rise is expected over the 21st century. In Vaasa, the decline will likely still continue for several decades before the turning point is reached by the end of the century. Uncertainties are large, however, as seen in the wide range of different percentiles plotted.

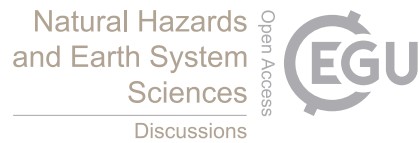

## 4  Discussion

We have presented probability distributions of mean sea level in 2100 along the Finnish coast for three emission scenarios. Differences between the scenarios are notable: Under the low emission scenario, the MSL change would likely remain negative over most of the Finnish coast during this century. Under the high emission pathway, a MSL rise is projected over the whole

coastline, and the 95th percentile projections reach 1.5 m in the Gulf of Finland. By coincidence, 1.5 m is also the height of the worst coastal flood on record in Helsinki, relative to the present-day mean sea level. The flood took place in January 2005 during the storm Gudrun and caused significant damage, which were alleviated with temporary flood protection structures. Even under the medium emission scenario, present-day worst-case coastal floods may become common by the end of the century (Pellikka et al., 2018), which would require substantial investments in adaptation.

The low emission scenario (RCP2.6/SSP1-2.6) represents ambitious climate mitigation in line with the Paris Agreement: global warming stays below 2°C as the emissions decline to net zero in the second half of the 21st century. In the medium scenario (RCP4.5/SSP2-4.5), emissions peak around mid-century, resulting in an estimated global warming of 2.7°C by 2100. The high scenario (RCP8.5/SSP5-8.5) represents a fossil-fuel intensive pathway, where emissions grow strongly during this century and warming by 2100 can exceed 4 °C (Chen et al., 2021).

IPCC does not assess the likelihood or feasibility of the different emission scenarios, but such estimates are often needed by policymakers, and implicit assessments of scenario probabilities are easily made by end users in the absence of relevant expert assessments (Ho et al., 2019). Hausfather and Peters (2020) argue that considering current trends in the energy sector, SSP5-8.5 should be regarded as a highly unlikely worst-case scenario, and that current policies point to the modest mitigation pathway SSP2-4.5. On the other hand, the feasibility of the low emission scenario can also be questioned, as the ambition in

climate change mitigation should strongly increase to keep the goals of the Paris Agreement – and hence SSP1-2.6 – within reach (den Elzen et al., 2022). This again points to SSP2-4.5 as the most likely scenario.

Our mixture distribution of GMSLR is in line with the IPCC AR6 projection (medium confidence) under the low emission scenario, but higher than that in the upper tail under medium and high emissions. The AR6 low confidence projection is similar to our mixture projection under medium emissions, and even higher under high emissions. It can be concluded that

AR6 manages to cover potential SLR futures better than earlier IPCC reports. In high risk applications, the low confidence projection should be considered in addition to the medium confidence projection.

The earlier published projections for the Finnish coast (Pellikka et al., 2018), used in national adaptation planning, are close to the medium-emission projections presented in this study. The probability distribution calculated for the high emission scenario shows potential for a considerably larger sea level rise, however. In contrast to earlier estimates, the expected absolute

SLR in Finland (excluding land uplift) is not lower than the global mean, as the reduction caused by glacier fingerprints is compensated by a larger-than-average ocean component (thermal expansion and ocean dynamics).

We have also examined past MSL trends on the Finnish coast. The apparent sea level trend observed at a tide gauge site in Finland is a combination of three trends: land uplift, regional sea level rise, and the trend in the wind-induced component





related to changes in the water balance of the semi-enclosed Baltic Sea. To illustrate the relative roles of all these components, we next discuss Helsinki as an example.

The observed trend in annual mean sea levels at Helsinki was $-1.88$ ($-0.37$) mm a$^{-1}$ in 1904–2018 (1993–2018). This consists of land uplift of $-3.69$ mm a$^{-1}$, regional sea level rise of 1.32 (3.15) mm a$^{-1}$, and the trend in the wind-induced component of 0.49 (0.17) mm a$^{-1}$. Thus, currently the trends of SLR and land uplift approximately balance each other. In particular, we note that the trend in the wind-induced component is large enough that excluding it from the analysis would lead to a significant bias in the estimates of regional SLR rates.

According to Gräwe et al. (2019) and Passaro et al. (2021), spatial variability in the absolute SLR trends is notable even within the Baltic Sea basin. Our results are lower than the absolute SLR trends calculated for the Estonian coast by Suursaar and Kall (2018) – the authors associate the difference between their SLR rates (2.0 to 2.8 mm a$^{-1}$ in 1900–2010) and GMSLR to poorly resolved local land subsidence and the meteorologically driven internal sea level dynamics of the Baltic Sea. Over the satellite altimetry era, our results are lower than the trends reported by Passaro et al. (2021) for the northern Baltic Sea based on satellite altimetry (up to 5 to 6 mm a$^{-1}$ in 1995–2019). Our SLR trend excludes the wind component (0.15 to 0.18 mm a$^{-1}$ in 1993–2018), which is included in the results of Passaro et al. (2021), but this does not explain the discrepancy comprehensively.

Over the 21st century, land uplift will continue at a constant rate, while significant acceleration in global sea level rise is expected together with a small increase in westerly winds in the northeast Atlantic, causing additional wind-induced sea level rise in the Baltic Sea. Due to accelerating SLR, the reverse of the historical declining MSL trend is imminent on the southern Finnish coast. North in the Bothnian Bay, postglacial land uplift is probably strong enough to compensate for SLR so that significant relative sea level rise is unlikely over this century, at least if carbon emissions are mitigated.

The SLR scenarios used in coastal planning need to be tailored individually to the planning project or context, depending on the level of risk tolerance. We have shown one possible method of making projections that span the full probability range and are applicable to different needs. The method can also be easily adapted for different purposes e.g. by adjusting the weights. However, the end users of all probabilistic SLR projections need to recognize that the probabilities are always partially subjective and dependent on different assumptions. As long as there is deep uncertainty related to ice sheet behaviour in the warming climate, it is recommended not to be overconfident in any seemingly precise SLR projection (Behar et al., 2017).

## 5   Conclusions

The main findings of this study can be summarized as follows:

1. Historical trends of absolute sea level rise on the Finnish coast – thus, excluding the effect of land uplift and wind-induced changes in Baltic sea level – are in accordance with global mean rates. This is expected to continue over the 21st century, too, as the smaller-than-average glacier contribution is compensated by the larger-than-average ocean contribution in the Baltic Sea.





2. Median projections of mean sea level change in Finland (2100 relative to 1995–2014) range from −43 to +16 cm for the low emission scenario, −28 to +31 cm for the medium scenario, and +1 to +61 cm for the high scenario depending on location. Lowest values are related to the area of strong land uplift in the north and highest to the Gulf of Finland in the south. The upper tail of the probability distribution characterizes the risk of higher sea level rise. The 95th percentiles range from −9 to +50 cm (low emissions), +24 to +83 cm (medium), and +93 to +152 cm (high) depending on location.

3. Our weighted combination of 10 individual GMSLR projections is close to or in between of the two probability distributions provided by IPCC AR6 (medium and low confidence). Thus, AR6 manages to cover the full spectrum of potential sea level rise by 2100 better than earlier IPCC reports, and other literature does not suggest fundamentally different projections. The low-confidence storyline should be considered in high-risk applications, such as nuclear power plant safety.

*Data availability.* Sea level data from the Finnish tide gauges is open data provided through the open data service of FMI (see https://en. ilmatieteenlaitos.fi/open-data) or the Permanent Service for Mean Sea Level (https://www.psmsl.org/). All data related to SLR projections, which is used in this study, is available from the referenced publications.



**Appendix: Additional results**

Here, we provide additional tables and figures that might be beneficial for the end users of this study. Table A1 shows the projections of zonal geostrophic wind from the individual CMIP5 models used in this study. Tables A2, A3, and A4 give the MSL projections for all Finnish tide gauges and different emission scenarios. Fig. A1 shows the individual probability
5 distributions that are used to compose the mixture distribution of GMSLR in this study.

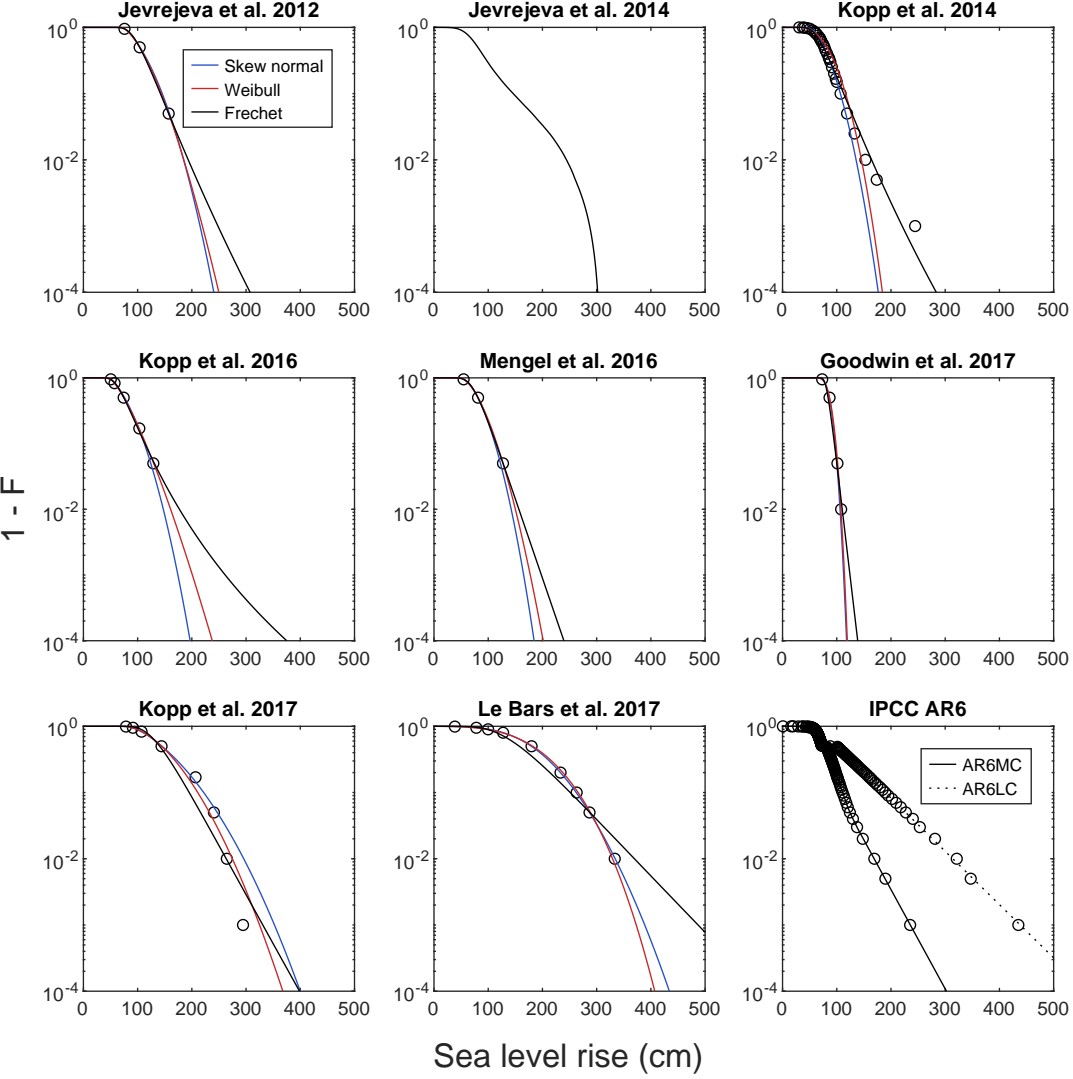

**Figure A1.** Members of the projection ensemble under the high emission scenario. Fits for all three distribution types (Weibull, Frechet, skew normal) are shown as complementary cumulative distribution functions (1 − F). No fits were applied to Jevrejeva et al. (2014) and IPCC AR6 (Fox-Kemper et al., 2021). AR6MC and AR6LC are the medium and low confidence projections of IPCC AR6, respectively.




**Table A1.** Twenty-year means of zonal geostrophic wind (metres per second) at 55°N, 15°E, according to CMIP5 model results: historical (1986–2005) and future (2081–2100) under low (RCP2.6), medium (RCP4.5), and high (RCP8.5) emission scenarios.

| Model | Hist. | Low | Medium | High |
|---|---|---|---|---|
| MIROC5 | 0.04 | −0.24 | 0.07 | 0.11 |
| MIROC-ESM | 2.97 | 3.38 | 3.85 | 4.55 |
| MIROC-ESM-CHEM | 3.01 | 3.42 | 3.86 | 4.54 |
| MRI-CGCM3 | 4.56 | 4.97 | 4.89 | 4.86 |
| BCC-CSM1-1 | 4.13 | 3.92 | 3.67 | 4.47 |
| NorESM1-M | 4.24 | 3.49 | 4.27 | 4.42 |
| HadGEM2-ES | 3.30 | 2.51 | 3.09 | 2.72 |
| MPI-ESM-LR | 3.13 | 3.56 | 3.51 | 4.31 |
| MPI-ESM-MR | 3.53 | 3.98 | 4.07 | 4.37 |
| CNRM-CM5 | 1.84 | 1.35 | 1.95 | 1.72 |
| IPSL-CM5A-LR | 3.06 | 2.27 | 3.30 | 4.22 |
| IPSL-CM5A-MR | 3.64 | 3.98 | 3.74 | 4.97 |
| GFDL-CM3 | 2.72 | 3.79 | 3.87 | 5.20 |
| GFDL-ESM2M | 2.09 | 3.07 | 3.23 | 3.79 |
| NCAR-CCSM4 | 4.86 | 5.35 | 4.80 | 5.96 |
| CanESM2 | 4.02 | 3.78 | 3.67 | 4.10 |
| EC-EARTH | 3.40 | 3.54 | 3.10 | 3.24 |

**Table A2.** Projected mean sea level (centimetres) at the Finnish tide gauges in 2100 under the low emission scenario (RCP2.6/SSP1-2.6).

| *Low scenario (2.6)* | 2100 (N2000) | | | | | 2100 relative to 1995–2014 | | | | |
|---|---|---|---|---|---|---|---|---|---|---|
| Station | 1% | 5% | 50% | 95% | 99% | 1% | 5% | 50% | 95% | 99% |
| Kemi | −50 | −43 | −20 | 14 | 42 | −70 | −62 | −40 | −5 | 22 |
| Oulu | −47 | −40 | −18 | 16 | 44 | −66 | −59 | −37 | −2 | 25 |
| Raahe | −53 | −45 | −23 | 12 | 39 | −71 | −63 | −41 | −7 | 21 |
| Pietarsaari | −55 | −48 | −26 | 9 | 37 | −73 | −66 | −43 | −9 | 19 |
| Vaasa | −53 | −46 | −24 | 10 | 38 | −71 | −64 | −42 | −8 | 20 |
| Kaskinen | −47 | −40 | −18 | 16 | 44 | −66 | −59 | −37 | −3 | 25 |
| Mäntyluoto | −39 | −32 | −10 | 24 | 52 | −57 | −50 | −28 | 6 | 34 |
| Rauma | −33 | −26 | −4 | 30 | 58 | −51 | −44 | −22 | 12 | 40 |
| Turku | −19 | −12 | 10 | 44 | 72 | −37 | −30 | −8 | 26 | 54 |
| Degerby | −25 | −18 | 4 | 38 | 66 | −40 | −33 | −11 | 23 | 51 |
| Hanko | −6 | 1 | 23 | 58 | 85 | −24 | −17 | 4 | 39 | 67 |
| Helsinki | 0 | 7 | 29 | 63 | 91 | −20 | −13 | 9 | 43 | 71 |
| Hamina | 8 | 15 | 37 | 72 | 100 | −14 | −7 | 16 | 50 | 78 |



**Table A3.** Projected mean sea level (centimetres) at the Finnish tide gauges in 2100 under the medium emission scenario (RCP4.5/SSP2-4.5).

| *Medium scenario (4.5)* | 2100 (N2000) | | | | | 2100 relative to 1995–2014 | | | | |
|---|---|---|---|---|---|---|---|---|---|---|
| Station | 1% | 5% | 50% | 95% | 99% | 1% | 5% | 50% | 95% | 99% |
| Kemi | −39 | −31 | −5 | 47 | 88 | −59 | −50 | −24 | 27 | 69 |
| Oulu | −37 | −28 | −2 | 49 | 91 | −56 | −47 | −21 | 30 | 72 |
| Raahe | −42 | −33 | −7 | 44 | 86 | −60 | −51 | −25 | 26 | 68 |
| Pietarsaari | −44 | −36 | −10 | 42 | 83 | −62 | −54 | −28 | 24 | 65 |
| Vaasa | −43 | −34 | −9 | 43 | 84 | −60 | −52 | −26 | 25 | 67 |
| Kaskinen | −37 | −28 | −3 | 49 | 90 | −56 | −47 | −21 | 30 | 71 |
| Mäntyluoto | −28 | −20 | 6 | 57 | 99 | −47 | −39 | −13 | 39 | 80 |
| Rauma | −22 | −14 | 12 | 63 | 105 | −40 | −32 | −6 | 45 | 86 |
| Turku | −9 | 0 | 25 | 77 | 118 | −26 | −18 | 7 | 59 | 100 |
| Degerby | −15 | −6 | 19 | 71 | 112 | −29 | −21 | 5 | 56 | 97 |
| Hanko | 5 | 13 | 39 | 90 | 132 | −14 | −6 | 20 | 72 | 113 |
| Helsinki | 11 | 19 | 45 | 96 | 138 | −9 | −1 | 25 | 76 | 118 |
| Hamina | 18 | 27 | 53 | 105 | 146 | −3 | 5 | 31 | 83 | 125 |

**Table A4.** Projected mean sea level (centimetres) at the Finnish tide gauges in 2100 under the high emission scenario (RCP8.5/SSP5-8.5).

| *High scenario (8.5)* | 2100 (N2000) | | | | | 2100 relative to 1995–2014 | | | | |
|---|---|---|---|---|---|---|---|---|---|---|
| Station | 1% | 5% | 50% | 95% | 99% | 1% | 5% | 50% | 95% | 99% |
| Kemi | −21 | −10 | 24 | 116 | 188 | −41 | −30 | 5 | 96 | 169 |
| Oulu | −19 | −8 | 27 | 118 | 190 | −38 | −27 | 8 | 99 | 171 |
| Raahe | −24 | −13 | 22 | 113 | 185 | −42 | −31 | 4 | 95 | 167 |
| Pietarsaari | −26 | −15 | 19 | 110 | 183 | −44 | −33 | 1 | 93 | 165 |
| Vaasa | −25 | −14 | 20 | 111 | 184 | −42 | −31 | 3 | 94 | 166 |
| Kaskinen | −19 | −8 | 26 | 118 | 190 | −38 | −27 | 7 | 99 | 171 |
| Mäntyluoto | −10 | 1 | 34 | 126 | 198 | −29 | −18 | 16 | 107 | 180 |
| Rauma | −4 | 6 | 40 | 132 | 204 | −23 | −12 | 22 | 114 | 186 |
| Turku | 9 | 20 | 54 | 145 | 218 | −9 | 2 | 36 | 128 | 200 |
| Degerby | 3 | 14 | 48 | 139 | 211 | −11 | −1 | 33 | 124 | 197 |
| Hanko | 23 | 34 | 68 | 159 | 231 | 4 | 15 | 49 | 140 | 212 |
| Helsinki | 29 | 40 | 74 | 165 | 237 | 9 | 20 | 54 | 145 | 217 |
| Hamina | 36 | 48 | 82 | 174 | 246 | 15 | 26 | 61 | 152 | 225 |





*Author contributions.* The analysis was performed by HP with the following contributions: KR analyzed the zonal geostrophic winds, MMJ the wind-induced sea level component and the historical trends, and MN extracted the land uplift rates. The manuscript was written by HP with contributions from all co-authors.

*Competing interests.* No competing interests are present.

5 *Acknowledgements.* This work was supported financially by VYR (National Nuclear Waste Management Fund) through SAFIR2018 (The Finnish Research Programme on Nuclear Power Plant Safety 2015–2018). The study has utilized research infrastructure facilities provided by FINMARI (Finnish Marine Research Infrastructure network). We wish to thank Dr. Jan-Victor Björkqvist for valuable comments and discussions, and Dr. Simo Siiriä for technical help.



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
