# Peer review of "Probabilistic projections and past trends of sea level rise in Finland"

_Natural Hazards and Earth System Sciences, 2022_

## Author Response (AR1)

**1 Authors' response to referee comments: nhess-2022-230**

We thank both referees for the constructive comments on our manuscript. Below, we have compiled
a list of all referee comments followed by our response (*in blue and italics*) and a short description of
the changes made in the manuscript (*in orange and italics*).

**7 Anonymous Referee #1**

This manuscript examines mean sea level trends along the Baltic coast of Finland, separating out
regional sea level rise, postglacial land uplift, and wind climate change effects. Tide gauge data and
empirical estimates of land uplift are utilised, along with numerical predictions of wind climate
change effects from climate models. The predicted local sea level rise trends approximately match
the established trend for global sea level rise. Mean sea level change probability distributions are
evaluated for three future emission pathway scenarios. It is found that the variation in mean sea
level change trends along the coast of Finland is particularly sensitive to postglacial uplift. The
introduction is comprehensive, relevant, and up to date. The research questions are clearly
articulated. In Table 1, the sea level rise projections from the early 2000s to 2100 highlight the
uncertainty in different model outputs, ranging from a minimum of 23 cm to a maximum of 287 cm!
The discussion is potentially very useful to coastal planners. The manuscript is written in a readable,
scientifically rigorous style, the results interpreted sensibly, and convincing findings deduced. The
manuscript is likely to be of interest to readers of Natural Hazards and Earth Sciences.

My recommendation is for acceptance of the manuscript once the following criticisms have been
addressed.

1.   It is worth commenting on numerical uncertainty in climate process models, even for the
same input data when solving the same equations with the same algorithms! Truncation and
round-off errors contribute to numerical noise that could reach the same order as the
solution (see e.g. S.J. Liao. On the reliability of computed chaotic solutions of non-linear
differential equations. Tellus A, 61(4): 550–564, 2009). I wonder how much this contributes
to the spread in process model projections.

*Response: In our understanding, this numerical uncertainty is not a very significant source of*
*error in climate projections. It is true that truncation and round-off errors accumulate when*
*simulating chaotic systems, such as weather, and lead to very different solutions depending*
*on numerical accuracy. However, climate models do not aim to simulate single weather*
*events, but statistical properties of weather over decades. Such long-term distributions are*
*much more stable and predictable than the day-to-day variability.*

*Various climate models do produce different results even under the same forcing. Partly this*
*stems from structural differences between models, partly from natural variability in the*
*climate system. This is why we have used a large ensemble of climate models (17 AOGCMs)*
*to project the geostrophic wind in the future. We have also used a wide range of SLR*
*projections from the literature to account for methodological differences and uncertainties.*

*Changes: A new paragraph discussing climate modelling uncertainties has been added at the*
*end of section 4.2 (page 19, lines 23–32 in the revised manuscript). This paragraph also deals*
*with comments 1–2 of Anonymous Referee #2.*

2.  Presumably, local changes in mean sea level in the Baltic have a knock-on effect on resonant
seiching within the semi-enclosed basin. Is this likely to be important in the future?

*Response: As the resonant periods of seiche oscillations depend on water depth, the change*
*in mean sea level does indeed have some effect on seiching. Numerical modelling would be*
*needed to quantify this impact in the study area, which is outside the scope of this paper. The*
*interaction between long-term and short-term sea level variations is an interesting question*
*also regarding other phenomena besides seiches – e.g. wind waves.*

*Changes: A new paragraph discussing future changes in short-term sea level variability has*
*been added at the end of section 4.4 (page 21, lines 1–6 in the revised manuscript).*

3.  Section 2.3. Is the mean sea level in the Baltic Sea affected by large-scale pressure variations
associated with teleconnections, such as the North Atlantic Oscillation? Please could the
authors comment upon this.

*Response: Yes, mean sea level in the Baltic Sea is substantially affected by the North Atlantic*
*Oscillation (NAO). The correlation between NAO and annual mean sea levels at the Finnish*
*coast have been studied in several papers listed below (Johansson et al. 2001, 2003, 2004).*
*The coefficients of determination ($R^2$) between detrended annual mean sea levels at the*
*Finnish coast and the normalized winter NAO index have been found to vary between 0.37–*
*0.46 depending on station (Johansson et al. 2003, 2004). High NAO is associated with a large*
*north–south air pressure difference over the North Atlantic, which in turn produces strong*
*westerly winds that tend to keep water level in the Baltic Sea basin high.*

*The sea level stations in the southern Baltic Sea show a weaker correlation with the NAO*
*index (Johansson et al. 2003) which the authors relate to the mean sea level slope within the*
*Baltic Sea. Westerly winds pile up water against the eastern coast of the Baltic Sea,*
*reinforcing the correlation between mean sea level and NAO at the Finnish coast.*

*In this paper, we use the zonal geostrophic wind $u_g$ in the southern Baltic Sea as the metric to*
*represent the variability in the large-scale atmospheric circulation. While it is related to the*
*same physical mechanism as the NAO index, namely the large-scale circulation over the*
*North Atlantic, the zonal geostrophic wind at this location has even higher correlations with*
*sea levels at the Finnish coast than the NAO index ($R^2$ = 0.84–0.89, Johansson et al. 2014).*

*Thus, the teleconnection associated with NAO is implicitly accounted for in our study, even*
*though we use a different metric that captures the effect even more closely than the NAO*
*index.*

*References:*

• *Johansson et al. 2001: Trends in sea level variability in the Baltic Sea. Boreal*
*Environment Research 6: 159–179.*
*http://www.borenv.net/BER/archive/pdfs/ber6/ber6-159s.pdf*
• *Johansson et al. 2003: An Improved Estimate for the Long-Term Mean Sea Level on*
*the Finnish Coast. Geophysica 39: 51–73.*
*https://www.geophysica.fi/pdf/geophysica_2003_39_1-2_051_johansson.pdf*
• *Johansson et al. 2004: Scenarios for sea level on the Finnish coast. Boreal*
*Environment Research 9: 153–166.*
*http://www.borenv.net/BER/archive/pdfs/ber9/ber9-153.pdf*

•   *Johansson et al. 2014: Global sea level rise scenarios adapted to the Finnish coast.*
*Journal of Marine Systems, 129: 35–46.*

*Changes: We have added an explanation of the connection between NAO and mean sea level*
*in the Baltic Sea on page 11, lines 10–13 in the revised manuscript.*

Minor corrections

1.   p1. L1. Change to "mean sea level (MSL) at the Finnish coast, in the northeastern Baltic Sea,
during the period 1901–2100."

*Corrected (P1, L1–2)*

2.   p3. L4. Delete "however,"

*Corrected (P3, L4)*

3.   p4. L7. Change to "What are the expected changes in mean sea level at the coast of Finland
by 2100?"

*Corrected (P4, L7)*

4.   p7 L19, p14 L6 and p21 Figure A1. Change "Frechet" to "Fréchet"

*All corrected.*

5.   p11. L5. Change to "… affect the dynamical sea level …"

*Corrected (P11, L5)*

6.   p13. Table 2. Change "metres per second" to "m/s" or "m.s-1".

*This unit is written out according to the journal instructions.*

7.   p14. Table 3. Change "millimetres per year" to "mm/yr" or "mm.yr-1".

*This unit is written out according to the journal instructions.*

8.   p15. L8. Change to "… if MICI projections are left out compared …"

*Corrected (P15, L9)*

9.   p15. L10. Change "controversality" to "controversy".

*Corrected (P15, L11)*

10. p16. Table 4. Change "centimetres" to "cm".

*This unit is written out according to the journal instructions.*

11. p16. Table 4. Change to "… to 2100 in the vicinity of some of the largest …"

*Corrected to "Projections of mean sea level change … for some of the largest coastal cities…"*
*(P17 Table 4)*

12. p17. L3. Change to "… a rise of 0.5 m is possible …"

*Corrected to "…a rise of 50 cm is possible…" (P16 L9)*

13. p18. L7. Change to "… during storm Gudrun and caused significant damage, which was
alleviated …"

*Corrected (P20 L30–31)*

14. p19. L29. Change to "Historical trends of absolute sea level rise on the Finnish coast,
excluding the effect of land uplift and wind-induced changes in Baltic sea level, are in
accordance with global mean rates."

*Corrected (P21 L9–10)*

15. p22. Table A1. Change "metres per second" to "m/s" or "m.s-1".

*This unit is written out according to the journal instructions.*

16. p22. Table A2. Change "centimetres" to "cm".

*This unit is written out according to the journal instructions.*

17. p23. Table A3. Change "centimetres" to "cm".

*This unit is written out according to the journal instructions.*

18. p24. Table A4. Change "centimetres" to "cm".

*This unit is written out according to the journal instructions.*

**Anonymous Referee #2**

This paper investigates past trends and future projections of mean sea level on the Finnish coast.
MSL change is divided into three components: regional sea level rise, land uplift and wind climate
changes. Land uplift rates are obtained from the semi-empirical model, which is independent of tide
gauge observations. This is an advance compared to previous studies. Tide gauge data and numerical
climate model are respectively used for estimating past and future projection of wind climate
change effects. In terms of past trends, local SLR after being subtracted the land uplift and wind
climate changes is approximately close to global trend. For future projection of SLR, an ensemble of
existing global projections is merged under a probability framework. Therefore, it yields probability
distributions of MSL change for low, medium and high emission scenario. Such a probability
distribution is very useful for policy makers and stakeholders. Also, it is revealed that spatial
variations in the MSL projections result essentially depends on the local land uplift rates. The
manuscript is well-written with comprehensive and up-to-date introduction, well-presented results
and convincing findings. Also, it is very timely to update the local projections after the publication of
AR6 and other recent studies. I believe this manuscript fits in very well with the scope of NHESS.

I would like to recommend the acceptance of this manuscript if the below concerns are
appropriately addressed.

1. I suppose the models for wind climate changes and land uplift are also subject to different
kinds of uncertainty. Please comment on the effect of such uncertainties on the final
projections.

*Response: The uncertainties in the final mean sea level projections are clearly dominated by*
*the uncertainty in global sea level rise projections. As seen from our Fig. 7, the effect of the*
*wind component is small compared to sea level rise and land uplift. The land uplift, on the*
*other hand, is of the same order of magnitude than sea level rise, but with much narrower*
*uncertainty ranges.*

*The land uplift has been observed for several decades in the region and thus, the observation*
*uncertainties are small and well known. The uncertainty from the GIA modelling part is less*
*well known, but the observations are constraining the GIA model output. As the total land*
*uplift model is a combination of the two parts (observations + GIA model), the uncertainties*
*become small. Typically, the computational uncertainty is an order of magnitude smaller*
*than the provided land uplift values (see Fig. 14 in Vestøl et al, 2019).*

*Changes: Please see text inserted on page 19, lines 23–32 in the revised manuscript.*

2. Why are the probability distributions for wind climate change and land uplift rates assumed
to be Gaussian? Any evidence to support this assumption? Have you ever tried any other
distributions? What are the effect of other distribution on the MSL probability distributions?

*Response: The Gaussian distribution is the simplest choice, and the one to be used if there is*
*no evidence to support some other type of distribution. The uncertainty in land uplift rates is*
*characterized by the standard deviation given by the NKG2016LU model, which we use to fit*
*the Gaussian distribution. Regarding the wind component, the uncertainty is characterized*
*by the output of the 17-model ensemble used to project the zonal geostrophic wind (Table 2).*
*For either process we do not have evidence that would point to a non-symmetrical*
*distribution.*

*In any case, as we comment above, the effect of uncertainties in the wind component and*
*land uplift is minor compared to the uncertainty in sea level rise projections. Therefore, there*
*would be little value in trying to elaborate the analysis of uncertainty distributions of the*
*wind component and land uplift.*

*Changes: Factors supporting the use of the Gaussian distribution are discussed in the revised*
*text on page 19, lines 23–32.*

3. Figure 7. This is a very useful graph, which supports the finding that "spatial variations in the
MSL projections result essentially depends on the local land uplift rates". However, I cannot
find enough clear description in the main text to interpret this graph.

*Response: Thank you for the comment.*

*Changes: We have added more explanation of the graph in the finalized manuscript (P15 L15*
*– P16 L2).*

4. The discussion. Indeed, before the discussion section, the manuscript is highly readable.
However, the discussion is not concise and streamlined. Reader like me can easily get lost. I
advised the authors to divide the discussion into several subsections regarding future
projections, past trends and spatial variability and etc.

*Response: Thank you for this comment which resulted in a clear improvement of the*
*manuscript.*

*Changes: We have rewritten the discussion and added subtitles.*

**Minor comments**

1.  Figure 7 a) The caption should be mean sea level change

*Corrected.*

2.  Figure 8. The vertical axis name should be mean sea level change according to the
description in main text. Please clarify.

*Corrected the description in the text on page 17, line 1 (the figure shows mean sea level in*
*the Finnish N2000 height system).*

3.  In Figs. 3 and 4, please add "low", "medium", "high" to the corresponding emission
scenarios to improve the readability of these graphs.

*Added.*